# Clinical Approach to the Patient in Critical State Following Immunotherapy and/or Stem Cell Transplantation: Guideline for the On-Call Physician

**DOI:** 10.3390/jcm8060884

**Published:** 2019-06-20

**Authors:** Catalin Constantinescu, Constantin Bodolea, Sergiu Pasca, Patric Teodorescu, Delia Dima, Ioana Rus, Tiberiu Tat, Patriciu Achimas-Cadariu, Alina Tanase, Ciprian Tomuleasa, Hermann Einsele

**Affiliations:** 1Department of Hematology, Iuliu Hatieganu University of Medicine and Pharmacy, 400124 Cluj Napoca, Romania; constantinescu.catalin@ymail.com (C.C.); pasca.sergiu123@gmail.com (S.P.); patric_te@yahoo.com (P.T.); 2Intensive Care Unit, Ion Chiricuta Clinical Cancer Center, 400005 Cluj Napoca, Romania; 3Department of Anesthesia, Iuliu Hatieganu University of Medicine and Pharmacy, 400124 Cluj Napoca, Romania; cbodolea@gmail.com; 4Department of Hematology, Ion Chiricuta Clinical Cancer Center, 400005 Cluj Napoca, Romania; deli_dima@yahoo.com (D.D.); codruta_21@yahoo.com (I.R.); dr_tibi@yahoo.com (T.T.); 5Department of Surgery, Iuliu Hatieganu University of Medicine and Pharmacy, 400124 Cluj Napoca, Romania; patric_te@yahoo.com; 6Department of Stem Cell Transplantation, Fundeni Clinical Institute, 022328 Bucharest, Romania; alinadanielatanase@yahoo.com; 7Research Center for Functional Genomics and Translational Medicine, Iuliu Hatieganu University of Medicine and Pharmacy, Victor Babes Street, 400124 Cluj Napoca, Romania; 8Department of Internal Medicine II, University Hospital Wuerzburg, 97070 Wuerzburg, Germany; Einsele_H@ukw.de

**Keywords:** hematology patient, MEWS-based clinical approach, ABCDE approach, sepsis screening, cytokine release syndrome

## Abstract

The initial management of the hematology patient in a critical state is crucial and poses a great challenge both for the hematologist and the intensive care unit (ICU) physician. After years of clinical practice, there is still a delay in the proper recognition and treatment of critical situations, which leads to late admission to the ICU. There is a much-needed systematic ABC (Airway, Breathing, Circulation) approach for the patients being treated on the wards as well as in the high dependency units because the underlying hematological disorder, as well as disease-related complications, have an increasing frequency. Focusing on score-based decision-making on the wards (Modified Early Warning Score (MEWS), together with Quick Sofa score), active sepsis screening with inflammation markers (C-reactive protein, procalcitonin, and presepsin), and assessment of microcirculation, organ perfusion, and oxygen supply by using paraclinical parameters from the ICU setting (lactate, central venous oxygen saturation (ScVO_2_), and venous-to-arterial carbon dioxide difference), hematologists can manage the immediate critical patient and improve the overall outcome.

## 1. Critically Ill Hematology Patients

The incidence of hematological malignancies has been increasing in the last few years. Modern approaches that include chemotherapy, radiotherapy, immunotherapy, and/or hematopoietic stem cell transplant (HSCT) lead to a higher number of patients that need further admission and treatment in an intensive care unit (ICU) [1,2,3,4,5]. The initial management of a patient diagnosed with a hematological malignancy in critical condition is crucial and poses a great challenge both for the hematologist and the ICU physician, because of the complex nature of the disease and the vast array of complications that arise. The main reasons for ICU admission are disease progression, organ dysfunctions, sepsis, and treatment-associated complications.

Despite advances in the management of such patients, hospitals still experience delayed admission or transfer to the ICU due to an improper assessment of critically ill patients. Thus, the current manuscript brings forward a kind reminder of the importance to train hematologists in the daily assessment of their patients and help recognize urgent scenarios that affect the overall survival (OS) and treatment-related mortality (TRM) of patients [6,7,8,9]. A standardized ABC (Airway, Breathing, Circulation) approach/algorithm is compulsory to improve both the diagnostic and treatment protocols. A daily joint assessment by the ward physician and ICU doctors for patients in need of a second opinion or triage is beneficial because it can trigger an early transfer to the ICU, where life-sustaining therapies could be started as soon as possible. Earlier ICU admission of patients with hematologic malignancies is associated with better survival, disease control, and a higher quality of life [10,11,12].

## 2. Acknowledgment of the Impending Critical Situation and Initial Management of the Critical Patient on the Ward: The Modified Early Warning Score (MEWS)

Hematologists must recognize and start treatment at the bedside. Thus, following HSCT, patients are usually treated in high dependency units, where they can be properly isolated and monitored, but physicians must pay constant attention to the changes in vital parameters [13,14,15]. An inappropriate approach with rushed clinical examination, over-burdened ward staff, and lack of medical training in recognizing impending organ failures are some of the reasons for delayed correct management [16]. Physical deterioration, with the most common premonitory signs being hypotension and a fall in the Glasgow Coma Scale/Score (GCS) (Table 1) [17,18,19,20], occurs over hours and if left unattended it impacts on patient morbidity and mortality [21,22]. Up to 25% of admissions or transfers to the ICU department from the ward have deteriorated to the point of cardiorespiratory arrest [23].

For the initial clinical evaluation, the Modified Early Warning Score (MEWS) (Table 2) has been the basic algorithm, already implemented in hospitals around the world and predicts patient outcomes, including cardiac arrest, ICU transfer, and in-hospital mortality [24]. MEWS score minimizes the risk of forgetting to evaluate some of the most important vital parameters, such as level of consciousness, respiratory rate, blood pressure, heart rate, and urine output. Nurses should be trained to use this score because it is important to assess the patient as soon as possible in order to decide further management. 

A value of at least 5 points should dictate the need for continuous monitoring of vital signs, ICU physician consult, and if required, transfer of the patient to a higher point of care [25].

## 3. ABCDE (Airway, Breathing, Circulation, Disability, Exposure) of the Potential Critically Ill Patient on the Ward

Following patient triage, physicians must use a stepwise approach to resuscitation management, as a rapid assessment of the patient and lifesaving interventions gain more time in critical situations, called ABCDE (Airway, Breathing, Circulation, Disability, Exposure) [26,27,28].

The first step is to assess the **consciousness** level by using the GCS. It uses verbal and painful stimuli. One must check for the presence of seizures. Also, before excluding any withdrawal syndromes, agitation and confusion might develop due to an organic dysfunction. Step 2 is **A—Airway** check of the patency of the airway. The physician can use chin lift and head tilt, or jaw thrust as airway opening maneuvers, and inspect the oropharynx for foreign bodies [29,30]. If the airway is clear and the patient is not breathing, step 3 is calling of the emergency team and starting basic life support (BLS). If possible, one should initiate bag-valve-mask ventilation in 100% inspired oxygen concentration (FiO_2_). If familiar with airway adjuncts, such as an oropharyngeal cannula or I-Gel, the hematologist can use them, but these techniques can be harmful in the hands of inappropriately trained staff. If the patient is breathing (**B—Breathing**), step 4 is inspection of the chest for movement, asymmetrical chest expansion, assessment of the respiratory rate, oxygen saturation (SpO_2_) and looking for cyanosis, wheezing, or stridor. Providing high oxygen concentration, by using a facial mask or reservoir mask, is helpful if beginning non-invasive ventilation to relieve hypoxemia is not possible. Thus, clinicians gain more time until ICU clinical management is optimal. 

**C—Circulation** refers to checking capillary refill time, heart rate, blood pressure, urine output, and skin temperature with differentiation between cold shock (hypovolemic/cardiogenic) or warm shock (septic/anaphylactic) [31,32]. Adequate intravenous access (at least one 18 gauge (G) intravenous catheter) is compulsory and if a shock state is suspected, one should begin fluid infusion (crystalloid or colloid) [33,34].

**D** stands for **Disability** and physicians must check again for Glasgow Coma Scale (GCS), blood glucose, and pupil diameter, whereas **E** stands for **Exposure** and temperature is checked. If sepsis is suspected, blood cultures are compulsory, and empiric antibiotic therapy must begin. Full exposure of the patient will facilitate physical examination [35,36]. Arterial blood gas (ABG) is useful to measure the adequacy of ventilation (PaCO_2_), oxygenation (PaO_2_, A-a gradient), and circulation (pH and lactate) and can guide response to treatment or it indicates further deterioration [37,38].

Help can alleviate the burden of the critical situation and can increase focus. As such, second medical support is of crucial importance. The ABCDE steps are depicted in Figure 1.

## 4. Screening for Sepsis: Quick Sequential Organ Failure (qSOFA) Score

Screening for sepsis in the hematology and transplant department should be done on a regular basis and care bundles started as soon as possible. Sepsis is defined as a life-threatening organ dysfunction caused by a dysregulated host response to infection, which can progress to septic shock [39,40,41]. Early sepsis detection improves outcomes and survival. Initially, sepsis was defined as documented or suspected infection and at least two signs of systemic inflammatory response syndrome (SIRS): Temperature >38.0 °C or <36.0 °C, heart rate (HR) > 90/min, respiratory rate (RR) > 20/min or PaCO_2_ < 4.3 kPa, white blood cell count (WBC) > 12.0 × 109/L or <4.0 × 109/L. Still, because the Systemic Inflammatory Response Syndrome (SIRS) criteria have poor specificity for sepsis [42], a recently developed promising risk stratification tool to facilitate easier identification of patients potentially at risk of dying from sepsis is the qSOFA (“quick sequential organ failure score”), presented along with the new sepsis definitions in the Surviving Sepsis Campaign [39]. The qSOFA score consists of three components: Altered mental status GCS < 15, systolic blood pressure less than or equal to 100 mmHg, and a respiratory rate of at least 22 breaths per minute. A score of 2 or higher had greater than 60% sensitivity for in-hospital mortality [43] (Table 3). The proposed use of qSOFA is to immediately identify high-risk infected patients outside the ICU and to prompt clinicians to consider additional diagnostic tests or escalation of therapy. Furthermore, the calculation of the complete Sequential (sepsis-related) Organ Failure Assessment (SOFA) score should be made, as should the Cellular Injury Score (CIS), which is an index of cellular injury, being calculated from three parameters of intracellular metabolism: Arterial ketone body ratio, osmolality gap, and blood lactate [44,45].

## 5. Procalcitonin, Presepsin, and C-Reactive Protein in the Immunotherapy and Transplant Setting

Infections play a major role in the management of a patient diagnosed with hematological malignancy which is under treatment with either chemo-immunotherapy, with/without HSCT. Patients with neutropenia after chemotherapy, as well as patients following HSCT, are especially predisposed to infections. Fever is frequent, but it is not a specific symptom of infections [46,47,48,49,50]. Biomarkers, such as procalcitonin (PCT), C-reactive protein (CRP), and presepsin, can aid in the diagnosis/exclusion of sepsis [51,52].

CRP is often used as a biomarker of an inflammatory syndrome and clinicians use it as guidance of therapy. Even if elevated CRP levels are often diagnosed in the clinic, infection is the most common cause [53,54]. PCT is the 116-amino acid prohormone molecule of calcitonin and is produced by the C cells of the thyroid gland as well as several other cell types and organs in response to proinflammatory stimulation. After PCT is secreted into the circulation, the peak plasma concentration reaches a stable plateau, with a half-life of 25 to 30 hours. High levels of PCT are associated with systemic bacterial, fungal, and protozoal infections, but not with viral infections [55,56,57,58].

Presepsin is a soluble CD14-subtype used as a new inflammatory biomarker, identified on the surface of monocytes and macrophages. Following infection, CD14 is phagocytized with pathogens and cleaved, with subsequently soluble CD14 subtype (presepsin) being generated and released. The molecule rises as early as within 2 h of inflammation onset, which is even earlier than PCT and CRP. Elevated presepsin levels in the peripheral blood have widely been used as a biomarker of sepsis, frequently used in the field of critical care medicine and emergency medicine [59,60,61]. 

PCT is a sensitive and specific marker of sepsis in patients with febrile neutropenia [62,63]. In a retrospective study carried out in Japan on patients under therapy for an underlying hematological malignancy, Ebihara et al. concluded that the levels of PCT, but not those of CRP or presepsin, were significantly higher in the infection group than in the uninfected group (*p* < 0.03), indicating that PCT might be a more sensitive biomarker of infections, with findings that presepsin might have less diagnostic value in patients with neutropenia [64]. In 2018, Baraka and Zakaria included patients with febrile neutropenia, usually following chemotherapy, and reported that plasma presepsin levels were elevated in neutropenic patients with bacterial infection. The sensitivity and specificity of presepsin is better than that of PCT and CRP for the identification of a subsequent bacterial infection [65]. Thus, a correlation between markers should help in further management in the never-ending search for a sepsis source.

## 6. Assessment of Microvascular Perfusion with Lactate, ScvO_2_, and Delta PCO_2_

Assessment of microvascular perfusion and guided resuscitation are of key importance, with common laboratory markers used by ICU physicians being peripheral lactate and central venous oxygen saturation (ScvO_2_) [66]. Lactate levels and ScvO_2_ levels are strongly linked to outcome [67,68]. These markers can be easily obtained on the ward.

Shock is defined as inappropriate oxygen supply–demand balance due to hypoperfusion, with the tissues increasing the extraction rate of oxygen, and as such maintaining the oxygen consumption. In time, the oxygen consumption drops, and lactate levels rise. The oxygen delivery (DO_2_) to tissues depends on cardiac output (CO), arterial oxygen content (CaO), and hemoglobinemia (Hb), so physicians should try to improve all of them to treat the shock state [69]. As a surrogate of mixed venous oxygen saturation (SvO_2_) for evaluating O_2_ demand/supply adequacy, central oxygen venous saturation (ScvO_2_) is a trustworthy parameter because it is more readily available. As it represents the amount of oxygen remaining in the systemic circulation after its passage through the tissues. ScvO_2_ correlates to the balance between DO_2_ and oxygen consumption (VO_2_) [70]. 

Most importantly, ScVO_2_ measurement can be obtained from blood drawn from a central venous catheter, either placed in the internal jugular vein or subclavian vein, which many hematologic patients have, where the tip is appropriately placed at the junction of the superior vena cava and right atrium. ScVO_2_ values less than 65% to 70% under acute patient conditions should alert us to the presence of tissue hypoxia or inadequate perfusion, which is associated with increased mortality in the general critical patient population. Abnormal ScVO_2_ values equal or greater than 90% suggest the presence of cryptic shock, because of poor oxygen utilization by the mitochondria and tissue oxygen concentration, and is also associated with increased mortality [71].

ScvO_2_ can also be used as a physiologic transfusion trigger. The studies conducted by Rivers et al. and Vallet et al. conclude that ScvO_2_ appears to be an interesting parameter to help with transfusion decisions in hemodynamically unstable patients with severe sepsis or in stable high-risk surgical patients equipped with a CVC. This suggestion merits a controlled randomized study [72,73].

Furthermore, ScvO_2_ < 70% could be considered a transfusion trigger although a lower ScvO_2_ (i.e., closer to 50%) would be more appropriate for identifying the anaerobic threshold [74]. Still, up to date, clinical data conducted on hematologic patients regarding the use of ScvO_2_ as a transfusion trigger has yet to be published.

The peripheral lactate value, obtainable from peripheral intravenous lines, is the end product (2-hydroxypropionic acid) of anaerobic glycolysis, a marker of secondary anaerobic metabolism, with normal values between 0.5 and 2 mmol/L, that has become one of the most widely used biomarkers of hypoperfusion [75]. A venous lactate value greater than or equal to 4 mmol/L has been used as an initial screen for sepsis-induced organ dysfunction. Still, a lactate threshold of 2 mmol/L has been recommended for the screening of sepsis-induced organ dysfunction, as according to the Third International Consensus for Sepsis and Septic Shock [76,77]. Thus, physicians should try to obtain clearance of lactate levels by initiating fluid resuscitation.

Another useful biochemical parameter is the venous-to-arterial carbon dioxide difference (delta PCO_2_), which is calculated by subtracting the arterial PCO_2_ from the central venous PCO_2_, also measured from a central vein placed catheter [78,79]. The pressure-controlled ventilation (Pcv)–aCO_2_ gap has been recently used as a resuscitation endpoint largely because CO_2_ is more soluble than O_2_ in the blood and may be able to more accurately detect microcirculatory dysfunction [80]. In multiple studies, a Pcv–aCO_2_ gap less than 6 mm Hg was associated with a higher cardiac index, better lactate clearance, and improved microcirculatory perfusion. Conversely, a Pcv-aCO_2_ gap that is greater than 6 mm Hg can reliably reflect critical hypoperfusion that may be amenable to fluid resuscitation with or without inotrope support [81]. Furthermore, the clinical interpretation of ScvO_2_ and Pcv–aCO_2_ gap and clinical approach to the critical patient on the ward are illustrated in Figure 2 and Figure 3.

## 7. Management of Cytokine Release Syndrome

Treatment of cancer with immune-based therapies represents a novel therapeutic approach [82,83,84]. As these therapies for cancer become potent, more effective, and more widely available, optimal management of their unique toxicities becomes increasingly important. Risks associated with cancer immunotherapy can be classified into autoimmune toxicity and cytokine-associated toxicity, also known as cytokine release syndrome (CRS), which can be found in the ward patient. Autoimmune toxicity results from an antigen-specific attack on host tissues. CRS is a massive cytokine release into the bloodstream following the administration of immune-based therapies, such as chimeric antigen receptor T cells (CAR-T cells) therapy, and other monoclonal antibodies, such as rituximab, blinatumomab, and alemtuzumab, and represents a life-threatening complication which requires early diagnosis and treatment because it can lead to multiorgan system failure in patients [85,86]. Its symptoms resemble very well that of sepsis or septic shock. CRS is associated with elevated circulating levels of several cytokines, including interleukin (IL)-6, interleukin-10 (IL-10), interferon-gamma (IFN-γ), tumor necrosis factor (TNF), interleukin-2 (IL-2), and interleukin-8 (IL-8). Emerging evidence implicates IL-6 as a central mediator of toxicity in CRS. IL-6 is an inflammatory cytokine involved in many processes, and is produced by monocytes/macrophages, dendritic cells, T cells, fibroblasts, keratinocytes, endothelial cells, adipocytes, myocytes, mesangial cells, and osteoblasts. Uncontrolled studies demonstrate that immunosuppression using anti-cytokine directed therapies, such as tocilizumab 8mg/kg intravenous once, an anti-IL-6 receptor antibody (IL-6R antagonist), with or without corticosteroids, can reverse the life-threatening CRS in patients treated with CART-19 or blinatumomab [87]. Patients predicted to develop severe CRS could be more closely monitored to allow early initiation of aggressive supportive care. The timing of symptom onset and CRS severity depends on the inducing agent and the magnitude of immune cell activation [88,89].

If anti-cytokine therapies fail, another possible method of improving the clinical symptoms could be the use of continuous renal replacement therapy (CRRT), which is widely used in septic shock with multiorgan failure [90]. The CRRT works as an artificial kidney, providing blood purification by using a semi-permeable membrane that uses diffusion, convection, and adsorption as physical principles. CRRT is a process of slow isotonic removal of water and solute, clearance of inflammatory mediators and myocardial depression factors, correction of electrolyte and acid–base imbalance, and maintenance of circulatory stability [91]. The efficiency of diffusive removal decreases with increasing molecular weight of the solute. Convective clearance is proportional to the ultrafiltration rate and independent of the molecular weight up to the cut-off point of the membrane, which is 30,000 to 50,000 Da for the currently used open hemofiltration membranes (IL-6 being 26 KDa). Adsorption to the membrane represents a third and for some solutes, an even more important route of elimination [92]. CRRT is only started in the ICU, as such it is important that an ICU physician should consult the patient on the ward, but to date, there is only one published case of CRS following CD-19 CAR-T-cell therapy successfully treated with CRRT [93].

The CytoSorb^®^ (Cytosorbents, Corporation, New Jersey, USA) device consists of a single use hemoadsorption cartridge, which can be used with standard blood pumps. The cartridge is filled with sorbent beads made from a porous polymer that adsorbs and captures cytokines as blood passes through the device. This process is concentration dependent, and so the higher the levels of cytokines in the blood, the faster the levels are reduced. The absorber has a surface of about 45,000 m^2^ compared to a conventional hemofilter with a surface of 1 to 1.5 m^2^, with a molecular cutoff of about 60 kDa, removing cytokines as well as other toxins and drugs. Multiple studies provide evidence of IL-6 cytokine removal [94,95]. There are no randomized controlled studies in this population.

As proof-of-concept, we analyzed the summary characteristics of hematological patients following either therapy with Blinatumomab or chimeric antigen receptor T cells (CAR-T cells). This has been performed through a literature review. The number of patients included in the CAR-T cell studies ranged from 30 to 53, while in the two studies in which Blinatumomab was used, the number of patients ranged from 20 to 271 (Figure 4) [96,97,98,99,100,101]. The median overall survival ranged from approximately 7.5 months to 13 months, with lower OS (overall survival) being observed in the study where Blinatumomab was used and OS was reported. The median event free survival ranged from approximately 6 to 8 months and was specified only in two studies where CAR-T cells were used. Considering the disease, one study included DLBCL (diffuse large B-cell lymphoma) patients, while the rest included ALL (acute lymphoblastic leukemia) patients. Both CRS and neurotoxicity were observed more frequently in patients treated with CAR-T cells. 

Neurotoxicity is an important and common complication of CAR therapies. Acute neurologic symptoms occur in many patients treated with CD19-directed CARs for B-cell malignancies [102]. Clinical manifestations include headache, confusion, delirium, language disturbance, seizures, and rarely, acute cerebral edema. Neurotoxicity is associated with increased cerebrospinal fluid (CSF) protein and white blood cell counts, correlated with increased migration of leukemia cells and serum proteins across the blood–CSF barrier. Although CAR-T cells are detected in the CSF of most, if not all, patients who undergo sampling during neurotoxicity, they can also be detected in the CSF of patients without neurotoxicity, suggesting that their presence in the CSF alone is not enough to induce neurologic side-effects [103].

The association of neurotoxicity with CRS is well recognized. Severe neurotoxicity is mainly seen in patients with concurrent or preceding CRS and is uncommon in the absence of CRS. Neurotoxicity appears to present later than CRS, may take longer to resolve, and is less responsive to IL-6-targeted therapies than CRS. Most investigators consider neurotoxicity to be a related, but distinct, entity from CRS. Patients with neurotoxicity also experience CRS, which may be treated with tocilizumab, an IL-6 receptor antibody, and/or corticosteroids to modulate T-cell activity, but the subsequent effect of these interventions on neurotoxicity is not well understood. Evidence for the efficacy of tocilizumab in neurotoxicity is limited to published data from phase I and II clinical trials, but additional studies are underway. Tocilizumab may be used to treat CRS or neurotoxicity in 23% to 48% of affected patients with acute lymphoblastic leukemia (ALL), 6% to 43% with non-Hodgkin’s lymphomas (NHL), and 0% to 25% with chronic lymphocytic leukemia (CLL), but the indication for treatment (CRS versus neurotoxicity) and outcomes of the intervention have not been consistently reported [104,105].

Dexamethasone and other corticosteroids are used as first line therapy for neurotoxicity. Dexamethasone has excellent CNS penetration and is a standard of care in the treatment of cerebral edema and inflammation in the setting of brain tumors and neurotrauma. High-dose methylprednisolone is usually reserved for more severe cases of CAR-T cell-associated neurotoxicity and is employed based on its well-studied safety profiles in neuroinfammatory disorders. Under investigation, some trials report the potential beneficial effect of siltuximab, a chimeric monoclonal antibody that directly binds IL-6, preventing it from binding with soluble and membrane-bound IL-6 receptors. It has been used to manage CRS and neurotoxicity, as well as to manage neurologic adverse events during CAR-T cell treatment of glioblastoma. The role of anti-seizure prophylaxis for patients receiving CAR-T cell immunotherapy has not clearly been determined. Some CAR-T cell investigators start all patients on anti-seizure prophylaxis, some use it only for patients with CRS, and others do not use seizure prophylaxis at all. Levetiracetam is frequently used for seizure prophylaxis, but the ideal timing, dose, and overall utility of this strategy are unknown. The adverse event profile is favorable, and it does not have the marked effects on drug metabolism observed with some other anti-seizure medications [102,106].

## 8. Ethical Concerns and Communication with Patients and Family Members

In recent years, progress made in life-sustaining therapies have impacted the survival of hematologic patients admitted to ICU as well. These also helped in improved evaluation of the chances of reversibility or unfavorable outcomes [107,108,109,110,111,112,113]. However, the mortality in these patients is still high, ranging from 33% to 58%, with a median of 40% [114,115,116,117,118]. Thus, it is of high importance to have a realistic assessment of the opportunity of admitting a hematologic patient to ICU, and also to have good and extensive communication with the patients and their families right from the beginning, and all through the collaboration.

Although the relevance of various scoring systems analyzing the risk of fast deterioration of patients is widely recognized, it is also important to admit that they do not always completely apply to the patient in the hematological setting [119,120,121]. Therefore, the hematology physician should discriminate, beforehand, between different situations that would justify, or not, the admission to the ICU. In those cases, in which the hematologic condition has reached a palliative stage or an end-stage with no other treatment options, the transfer should not be an option. On the other hand, patients who are under a first-line treatment, patients who reached complete remission, or have a good response to the ongoing treatment should be, nevertheless, proposed for intensive care if the situation requires it. Other justified candidates would be those patients who showed only partial responses, have chemosensitive relapses, or have reasonable chances to obtain a response with further-line treatments. Last, but not least, patients undergoing experimental treatments, or treatments with high-risk mortality or known severe side-effects, could also benefit from admission to the ICU if, again, their status requires it [122].

All these matters that need to be considered should be evaluated and planned in advance through a consistent collaboration between the hematology and the intensive-care physicians. Also, these should be thoroughly explained to the patients and to their families, which should be part of the advance care planning [123,124,125].

Patients and their families should be taught right from the beginning about the possible life-threatening evolution, and the patient’s opinion should be asked for before care planning. Afterward, the transfer-decision should be the result of an evaluation made by a team of hematology and intensive-care physicians, instead of a one-sided view, in order to decide the best option for the patient. Once admitted to the ICU, the patient should be regularly re-evaluated by the hematology physician as well, who should continue to be part of the decision-making process [122].

During the admission to the ICU, the evolution of the patients could be variable. If the status improves, life-sustaining therapy should be continued, until retransfer to the hematology unit. Otherwise, if it deteriorates further, the option of withholding therapy should be discussed [115,125]. The third scenario is that in which the status remains unchanged, despite the life-sustaining therapies. In this case, the therapies should be continued, but the family should be informed that further deterioration should not necessarily involve the addition of an increase of therapies [126,127]. 

Any of these situations should be extensively and rigorously explained to the patients and their relatives, with any decision having the ultimate purpose of offering the best comfort for the patients and support for the families.

## 9. Conclusions

Clinical management in critical hematological patients should focus on score-based decision-making (MEWS, qSOFA, GCS), the role of biomarkers (CRP, PCT, presepsin), monitoring of microcirculation endpoints, organ perfusion, oxygen supply, achieving respiratory and hemodynamic stability, and macrocirculatory goals (including adequate intravascular volume resuscitation, improving cardiac output, assessing urine output). By using a step by step approach and exercising it, physicians will acquire the capabilities and skills to make quick decisions when challenging situations arise. Concerning ethical dilemmas, dealing with illness in the final stage raises questions about medical decisions. Effective communication with the patient and relatives, involving the family in the decision-making, and seeking ICU physician support is of the utmost importance in the search for improving the quality of life of our patients.

It might take years until health care professionals will assimilate and correctly apply the information provided. We encourage the training of hematologists and nurses in critical situations and better communication between health care providers with the final goal of decreasing the morbidity and mortality in this population and reaching a standard of care.

## Figures and Tables

**Figure 1 jcm-08-00884-f001:**
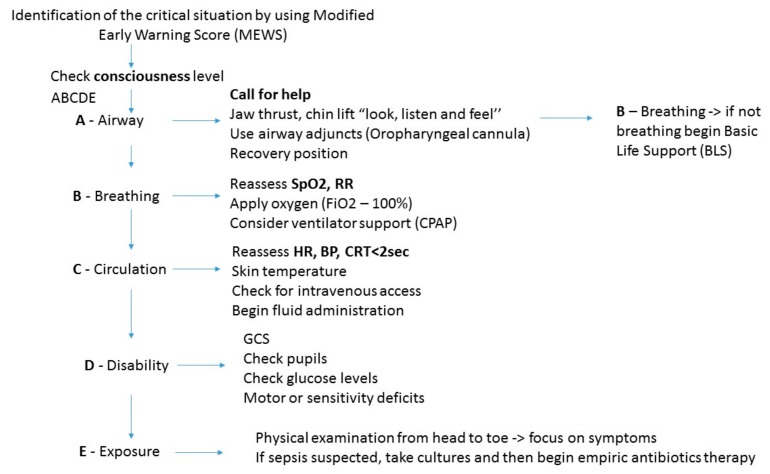
ABCDE (Airway, Breathing, Circulation, Disability, Exposure) steps. SpO_2_—peripheral oxygen saturation; RR—respiratory rate; CPAP—continuous positive airway pressure; HR—heart rate; BP—blood pressure; CRT—capillary refill time; GCS—Glasgow Coma Scale.

**Figure 2 jcm-08-00884-f002:**
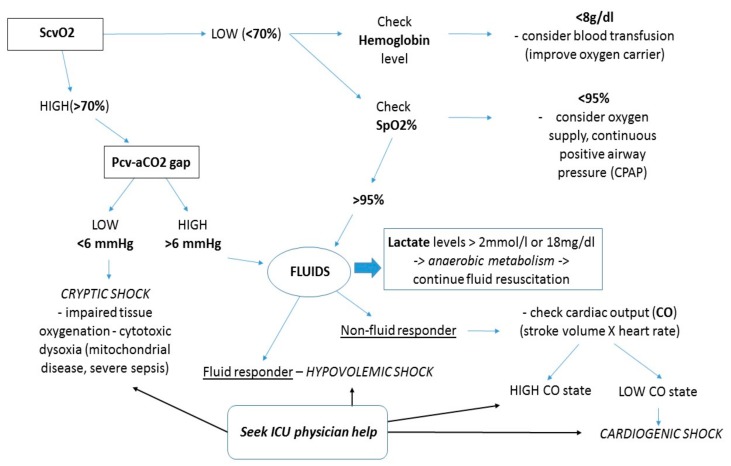
Clinical interpretation of ScvO_2_ and Pcv–aCO_2_ gap.

**Figure 3 jcm-08-00884-f003:**
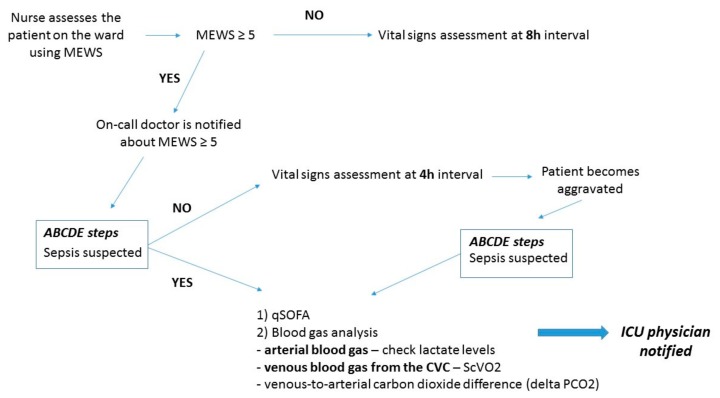
Clinical approach to the critical patient on the ward.

**Figure 4 jcm-08-00884-f004:**
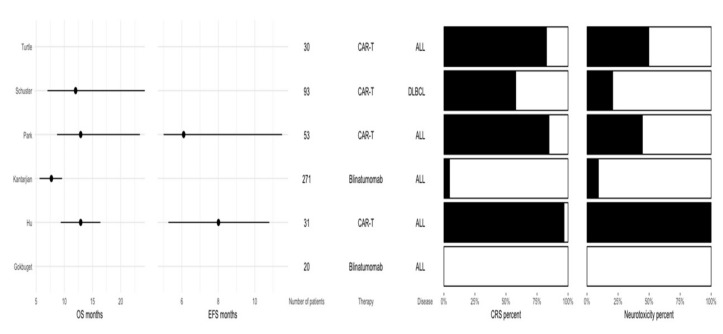
Summary of studies using either Blinatumomab or CAR-T cells as a therapeutic approach.

**Table 1 jcm-08-00884-t001:** Glasgow Coma Scale/Score (GCS).

Best Eye Response	Best Verbal Response	Best Motor Response
To pain (+2)	Oriented (+5)	Obeys commands (+6)
To verbal command (+3)	Confused (+4)	Localizes pain (+5)
Spontaneously (+4)	Inappropriate words (+3)	Withdrawal from pain (+4)
No eye opening (+1)	Incomprehensible sounds (+2)	Flexion to pain (+3)
Not assessable (trauma, edema) (+1c)	No verbal response (+1)	Extension to pain (+2)
	Intubated (+1t)	No motor response (+1)

**Table 2 jcm-08-00884-t002:** Modified Early Warning Score (MEWS).

Vital Parameters	3	2	1	0	1	2	3
Respiratory rate		≤8		9–14	15–20	21–19	>30
SpO_2_	≤91	92–93	94–95	≥96			
Temperature		≤35.0	35.1–36.0	36.1–38.0	38.1–38.5	>38.6	
Systolic blood pressure	<70	71–80	81–100	101–199		>200	
Heart rate		<40	40–50	51–100	101–110	111–129	>129
Level of consciousness	U = GCS 6	P = GCS 8	V = GCS 13	A = GCS 15			
Urine output (measured hourly)	<10 mL/h	<30 mLl/h	<45 mL/h				

U—unconscious; P—responds to pain; V—responds to voice; A—alert; GCS—Glasgow Coma Scale

**Table 3 jcm-08-00884-t003:** Quick Sequential Organ Failure Score (qSOFA).

Quick SOFA Score (qSOFA)
Altered mental status GCS < 15
Respiratory rate ≥ 22
Systolic BP ≤ 100 mmHg

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
