# Peer review of "Clinical Approach to the Patient in Critical State Following Immunotherapy and/or Stem Cell Transplantation: Guideline for the On-Call Physician"

_jcm, 2019, doi:10.3390/jcm8060884_

Round 1

Reviewer 1 Report

Dear Authors,

Apart from english language editing, I suggest to adda section on CAR-T related neurotoxicity/ICANS. I also suggest to add stepwise management of neutropenic fever/sepsis.

Author Response

            Dear Editor,

              Thank you very much for reviewing our manuscript. We appreciate the tremendous effort and time the reviewers devoted to improving our manuscript. We sincerely feel that their thoughtful comments have further strengthened the manuscript. Specific responses to each comment are presented in the Responses to the Reviewers. In the revised manuscript, revisions to the manuscript are indicated in red font. We hope that our responses to the reviewers’ comments and the revisions made to the manuscript satisfy all questions and concerns. 

               With my best regards,

             Ciprian Tomuleasa, M.D.

             Department of Hematology,

             Iuliu Hatieganu University of Medicine and Pharmacy, Cluj Napoca, Romania.

 Comments to the Editor

 We have answered all the very constructive criticism of the three reviewers and are confident that our manuscript is better and brings a new and useful insight for both hematologists and intensivists regarding the interdisciplinary approach for the management of the critically ill patient diagnosed with a hematological malignancy. For the revised manuscript, professors Daniela Ionescu and Mihnea Zdrenghea significantly contributed and thus, were added as author of the manuscript.

Comments to the Reviewers

 Reviewer #2:

            Comments and Suggestions for Authors

Superficial review of critical care in the hematology patient. The review adds little truly useful information to assist a practising hematologist. The scoring systems are simplistic and capture routine observations recording on the nursing chart - deterioriation in these leads to "MET" calls and ICU/HDU involvement anyway. The sections on sepsis and CRS are also overly superficial.

            Thank you very much for an important feed-back. The scoring systems, even they are simplistic, represent very important details in read-world clinical management of these patients. We consider that these very simple solutions are of great help to the practicing hematologist and disagree that they add truly useful information. The manuscript is written by practicing hematologists in collaboration with practicing ICU physicians and is intended to act as a bridge between the two specialties. It offers a clear-cut approach to the management of the critically ill patient, that impact the outcome of therapy. We have addressed the very constructive critiques and have rephrased some of the paragraphs, as well as added new one, in red in the revised manuscript. Thus, the abstract was restructured and re-written, which now is “The initial management of the hematology patient in critical state is crucial and poses a great challenge both for the hematologist and the intensive care unit (ICU) physician. After years of clinical practice, there is still a delay in the proper recognition and treatment of critical situations which leads to late admission to the ICU. There is a much-needed systematic ABC approach for the patients being treated on the wards as well as in the high dependency units because the underlying hematological disorder, as well as disease-related complications, have an increasing frequency. Focusing on score-based decision-making on the wards (Modified Early Warning Score (MEWS), together with Quick Sofa score), active sepsis screening with inflammation markers (C-reactive protein, procalcitonin and presepsin), and assessing microcirculation, organ perfusion and oxygen supply by using paraclinical parameters from the ICU setting (lactate, central venous oxygen saturation (ScVO2) and venous-to-arterial carbon dioxide difference), hematologists can manage the immediate critical patient and improve the overall outcome. “.

            In the first chapter, we have added that “A standardized ABC approach/algorithm is compulsory in order to improve both the diagnostic and treatment protocols. A daily joint assessment of the ward physician and ICU doctors for patients in need for a second opinion or triage is beneficial because it can trigger an early transfer to the ICU where life-sustaining therapies could be started as soon as possible.”, as well as “For the initial clinical evaluation, the Modified Early Warning Score (MEWS) has been the basic algorithm, already implemented in hospitals around the world and predicts patient outcomes, including cardiac arrest, ICU transfer and in-hospital mortality. MEWS score minimizes the risk of forgetting to evaluate some of the most important vital parameters, such as level of consciousness, respiratory rate, blood pressure, heart rate, and urine output. Nurses should be trained to use this score because it is important to assess the patient as soon as possible in order to decide further management.“.

            Chapter 5 was re-written, which now is “Infections play a major role in the management of a patient diagnosed with hematological malignancy which is under treatment with either chemo-immunotherapy, with/without HSCT. Patients with neutropenia after chemotherapy, as well as patients following HSCT, are especially predisposed to infections. Fever is frequent, but it is not a specific symptom of infections. Biomarkers such as procalcitonin (PCT), C-reactive protein (CRP) and presepsin can aid in the diagnosis/exclusion of sepsis.

CRP is often used as a biomarker of an inflammatory syndrome and clinicians use it as guidance of therapy. Even if elevated CRP levels are often diagnosed in the clinic, infection is the most common cause. PCT is the 116-amino acid prohormone molecule of calcitonin and is produced by the C cells of the thyroid gland as well as several other cell types and organs in response to proinflammatory stimulation. After PCT is secreted into the circulation, the peak plasma concentration goes to a stable plateau, with a half-life of 25–30 hours. High levels of PCT are associated with systemic bacterial, fungal, and protozoal infections, but not with viral infections. Presepsin is a soluble CD14-subtype used as a new inflammatory biomarker, identified on the surface of monocytes and macrophages. Following infection, CD14 is phagocytized with pathogens and cleaved, with subsequently soluble CD14 subtype (presepsin) being generated and released. The molecule rises as early as within 2 hours of inflammation onset, which is even earlier than PCT and CRP. Elevated presepsin levels in the peripheral blood have widely been used as a biomarker of sepsis, frequently used in the field of critical care medicine and emergency medicine. PCT is a sensitive and specific marker of sepsis in patients with febrile neutropenia. In a retrospective study carried out in Japan on patients under therapy for an underlying hematological malignancy, Ebihara et al have concluded that the levels of PCT, but not those of CRP or presepsin, were significantly higher in the infection group than in the uninfected group (P<0.03), indicating that PCT might be a more sensitive biomarker of infections, with findings that presepsin might have less diagnostic value in patients with neutropenia [64]. In 2018, Baraka and Zakaria have included patients with febrile neutropenia, usually following chemotherapy, and reported that plasma presepsin levels were elevated in neutropenic patients with bacterial infection. The sensitivity and specificity of presepsin are better than that of PCT and CRP for the identification of a subsequent bacterial infection [65]. Thus, a correlation between markers should help in further management in the never-ending search for a sepsis source.“.

We have also added that “ScvO2 can also be used as physiologic transfusion trigger. The studies conducted by Rivers et al. and Vallet et al. conclude that ScVO2 appears to be an interesting parameter to help with transfusion decisions in hemodynamically unstable patients with severe sepsis or in stable high-risk surgical patients equipped with a CVC. This suggestion merits a controlled randomized study. ScvO2 <70% could be considered a transfusion trigger although a lower ScvO2 (i.e., closer to 50%) would be more appropriate for identifying the anaerobic threshold. Still, up to date, clinical data conducted on hematologic patients regarding the use of ScvO2 as a transfusion trigger has yet to be published. “. These patients are at risk of developing cytokine release syndrome, a condition that is sometimes refractory to conventional therapy. For these cases, continuous renal replacement therapy (CCRP) is a valid alternative and thus we have added in the revised manuscript that “ CRRT is only started in the ICU, as such it is important that an ICU physician should consult the patient on the ward, but to date, there is only one case of CRS following CD-19 CAR-T-Cell therapy successfully treated with CRRT. CytoSorb® (Cytosorbents, Corporation, New Jersey, USA) device consists of a single use hemoadsorption cartridge which can be used with standard blood pumps. The cartridge is filled with sorbent beads made from a porous polymer that adsorbs and capture cytokines as blood passes through the device. This process is concentration dependent, and so the higher the levels of cytokines in the blood, the faster the levels are reduced. The absorber has a surface of about 45,000 m2 compared to a conventional hemofilter with a surface of 1–1.5 m2 with a molecular cutoff of about 60 kDa removing cytokines as well as other toxins and drugs. Multiple studies provide evidence of IL-6 cytokine removal. There are no randomized controlled studies in this population. “.

We have also added a new chapter, with focus on ethics and communication with the patient and it’s family, which now is “Ethical concerns and communication with patients and family members. In recent years, progresses made in life-sustaining therapies have impacted the survival of hematologic patients admitted to ICU as well. These also helped in better evaluating the chances of reversibility or unfavorable outcomes. However, the mortality in these patients is still high, ranging from 33% to 58%, with a median of 40%. Thus, it is of high importance to have a realistic assessment of the opportunity of admitting a hematologic patient to ICU, and also to have good and extensive communication with the patients and their families right from the beginning, and all through the collaboration.

Although the relevance of various scoring systems analyzing the risk of fast deterioration of patients is widely recognized, it is also important to admit that they do not always completely apply to the patient in the hematological setting. Therefore, the hematology physician should discriminate, beforehand, between different situations that would justify, or not, the admission to the ICU. In those cases, in which the hematologic condition has reached a palliative stage or an end-stage with no other treatment options, the transfer should not be an option. On the other hand, patients who are under a first-line treatment, patients who reached complete remission, or have a good response to the ongoing treatment should be nevertheless proposed for intensive care if the situation requires it. Other justified candidates would be those patients who showed only partial responses, have chemosensitive relapses or have reasonable chances to obtain a response with further-line treatments. Last, but not least, patients undergoing experimental treatments, or treatments with high-risk mortality or known severe side-effects, could also benefit from admission to the ICU if, again, their status requires it.

All of these matters that need to be considered, should be evaluated and planned in advance through a consistent collaboration between the hematology and the intensive-care physicians. Also, these should be thoroughly explained to the patients and to their families which should be part of the advance care planning.

The patients and the families should be taught right from the beginning about the possible life-threatening evolution, and the patients’ opinion should be asked for before the care planning. Afterward, the transfer-decision should be the result of an evaluation made by a team of hematology and intensive-care physicians, instead of a one-sided view, in order to decide the best option for the patient. Once admitted to the ICU, the patient should be regularly re-evaluated by the hematology physician as well, who should continue to be part of the decision-making process. During the admission to the ICU, the evolution of the patients could be variable. If the status improves, the life-sustaining therapy should be continued, until retransfer to the hematology unit. Otherwise, if it deteriorates furthermore, the option of withholding the therapies should be discussed. The third scenario is that in which the status remains unchanged, despite the life-sustaining therapies. In this case, the therapies should be continued, but the family should be informed that further deterioration, should not necessarily involve the addition of an increase of therapies.

Any of these situations should be extensively and rigorously explained to the patients and their relatives, any decision having an ultimate purpose to offer the best comfort for the patients, and support for the families.

The conclusion was also rephrased, which now is “Conclusion. Clinical management in critical hematological patients should focus on score-based decision-making (MEWS, qSOFA, GCS), the role of biomarkers (CRP, PCT, presepsin), monitoring of microcirculation endpoints, organ perfusion, oxygen supply, achieving respiratory and hemodynamic stability, macrocirculatory goals (including adequate intravascular volume resuscitation, improving cardiac output, and assessing urine output. It might take years until the health care professionals will assimilate and apply the information provided. We encourage improving the training of hematologists and nurses in critical situations, better communication between health care providers with the final goal of reaching a standard of care. “

Reviewer 2 Report

Superficial review of critical care in the hematology patient. The review adds little truly useful information to assist a practising hematologist. The scoring systems are simplistic and capture routine observations recording on the nursing chart - deterioriation in these leads to "MET" calls and ICU/HDU involvement anyway. The sections on sepsis and CRS are also overly superficial

Author Response

(The authors gave the same response as above.)

Reviewer 3 Report

The authors intend to describe a clinical approach to patients who need critical care after immunotherapy and/or stem cell therapy. They focus on score-based decision-making (MEWS, qSOFA, GCS), the role of biomarkers (CRP, PCT, presepsin) and monitoring of microcirculation, perfusion and oxygen supply.  

Topics and recommendations are less specific for patients after immunotherapy or stem cell transplantation. Graft versus host disease, non-infectious pneumonitis, immune-related adverse events or CAR-T cell-related encephalitis represent complications that are more specific but are not or only vaguely discussed. The role of scoring systems for is very controversial in this population. Communication with patients and family members about impact of intensive care on prognosis and further cancer treatment is of crucial importance but not mentioned at all.

Further explanations are required for:

Rows 69/70: “..Therefore, it is important to assess the patient as soon as possible, decide if there is need for urgent initiation of chemotherapy or plasmapheresis...”

Row 190: “..ScvO2 can also be used as a red blood cell transfusion trigger..” Is this really a clinical practice?

Rows 246ff and Figure 3 (what is Table 4?): “..systematic review of the literature..”

Author Response

            Dear Editor,

              Thank you very much for reviewing our manuscript. We appreciate the tremendous effort and time the reviewers devoted to improving our manuscript. We sincerely feel that their thoughtful comments have further strengthened the manuscript. Specific responses to each comment are presented in the Responses to the Reviewers. In the revised manuscript, revisions to the manuscript are indicated in red font. We hope that our responses to the reviewers’ comments and the revisions made to the manuscript satisfy all questions and concerns. 

               With my best regards,      

            Ciprian Tomuleasa, M.D.

             Department of Hematology,

             Iuliu Hatieganu University of Medicine and Pharmacy, Cluj Napoca, Romania.       

Comments to the Editor

We have answered all the very constructive criticism of the three reviewers and are confident that our manuscript is better and brings a new and useful insight for both hematologists and intensivists regarding the interdisciplinary approach for the management of the critically ill patient diagnosed with a hematological malignancy. For the revised manuscript, professors Daniela Ionescu and Mihnea Zdrenghea significantly contributed and thus, were added as author of the manuscript.

Comments to the Reviewers

 Reviewer #3:

The authors intend to describe a clinical approach to patients who need critical care after immunotherapy and/or stem cell therapy. They focus on score-based decision-making (MEWS, qSOFA, GCS), the role of biomarkers (CRP, PCT, presepsin) and monitoring of microcirculation, perfusion and oxygen supply.  

Topics and recommendations are less specific for patients after immunotherapy or stem cell transplantation. Graft versus host disease, non-infectious pneumonitis, immune-related adverse events or CAR-T cell-related encephalitis represent complications that are more specific but are not or only vaguely discussed. The role of scoring systems for is very controversial in this population. Communication with patients and family members about impact of intensive care on prognosis and further cancer treatment is of crucial importance but not mentioned at all.

Thank you very much for an important feed-back. We have addressed all the suggestions in red in the revised manuscript. We have also added in red in the manuscript a chapter on ethical concerns and communication with the patients and family members:

“8. Ethical concerns and communication with patients and family members

In recent years, progresses made in life-sustaining therapies have impacted the survival of hematologic patients admitted to ICU as well. These also helped in better evaluating the chances of reversibility or unfavorable outcomes. However, the mortality in these patients is still high, ranging from 33% to 58%, with a median of 40%. Thus, it is of high importance to have a realistic assessment of the opportunity of admitting a hematologic patient to ICU, and also to have good and extensive communication with the patients and their families right from the beginning, and all through the collaboration.

Although the relevance of various scoring systems analyzing the risk of fast deterioration of patients is widely recognized, it is also important to admit that they do not always completely apply to the patient in the hematological setting. Therefore, the hematology physician should discriminate, beforehand, between different situations that would justify, or not, the admission to the ICU. In those cases, in which the hematologic condition has reached a palliative stage or an end-stage with no other treatment options, the transfer should not be an option. On the other hand, patients who are under a first-line treatment, patients who reached complete remission, or have a good response to the ongoing treatment should be nevertheless proposed for intensive care if the situation requires it. Other justified candidates would be those patients who showed only partial responses, have chemosensitive relapses or have reasonable chances to obtain a response with further-line treatments. Last, but not least, patients undergoing experimental treatments, or treatments with high-risk mortality or known severe side-effects, could also benefit from admission to the ICU if, again, their status requires it.

All of these matters that need to be considered, should be evaluated and planned in advance through a consistent collaboration between the hematology and the intensive-care physicians. Also, these should be thoroughly explained to the patients and to their families which should be part of the advance care planning.

The patients and the families should be taught right from the beginning about the possible life-threatening evolution, and the patients’ opinion should be asked for before the care planning. Afterward, the transfer-decision should be the result of an evaluation made by a team of hematology and intensive-care physicians, instead of a one-sided view, in order to decide the best option for the patient. Once admitted to the ICU, the patient should be regularly re-evaluated by the hematology physician as well, who should continue to be part of the decision-making process.

During the admission to the ICU, the evolution of the patients could be variable. If the status improves, the life-sustaining therapy should be continued, until retransfer to the hematology unit. Otherwise, if it deteriorates furthermore, the option of withholding the therapies should be discussed. The third scenario is that in which the status remains unchanged, despite the life-sustaining therapies. In this case, the therapies should be continued, but the family should be informed that further deterioration, should not necessarily involve the addition of an increase of therapies.

Any of these situations should be extensively and rigorously explained to the patients and their relatives, any decision having an ultimate purpose to offer the best comfort for the patients, and support for the families. “.

-        Further explanations are required for:

Rows 69/70: “..Therefore, it is important to assess the patient as soon as possible, decide if there is need for urgent initiation of chemotherapy or plasmapheresis...”

Thank you very much for an important feed-back. We consider that this paragraph is redundant in the revised manuscript and have deleted it.

Row 190: “..ScvO2 can also be used as a red blood cell transfusion trigger..” Is this really a clinical practice?

Thank you very much for an important feed-back. We have added in red in the manuscript that “ScvO2 can also be used as physiologic transfusion trigger. The studies conducted by Rivers et al. and Vallet et al. conclude that ScVO2 appears to be an interesting parameter to help with transfusion decisions in hemodynamically unstable patients with severe sepsis or in stable high-risk surgical patients equipped with a CVC. This suggestion merits a controlled randomized study.

 ScvO2 <70% could be considered a transfusion trigger although a lower ScvO2 (i.e., closer to 50%) would be more appropriate for identifying the anaerobic threshold. Still, up to date, clinical data conducted on hematologic patients regarding the use of ScvO2 as a transfusion trigger has yet to be published. “.

Rows 246ff and Figure 3 (what is Table 4?): “..systematic review of the literature..”

Thank you very much for an important feed-back. In this paragraph we indeed present the main published side-effects following CAR T cell or blinatumomab administration. A separate systematic review is beyond the purpose of the manuscript and we chose to present the most important side-effects in a separate paragraph, as a proof-of-concept. We have rephrased the paragraph in red in the revised manuscript, which now is: “As proof-of-concept, we analyzed the summary characteristics of hematological patients following either therapy with Blinatumomab or CAR-T cells. This has been performed through a literature review. The number of patients included in the CAR-T cell studies ranged from 30 to 53, while in the two studies in which Blinatumomab was used, the number of patients ranged from 20 to 271 (Figure 3). The median overall survival ranged from approximately 7.5 months to 13 months, with lower OS being observed in the study where Blinatumomab was used and OS was reported. The median event free survival ranged from approximately 6 to 8 months and was specified only in two studies where CAR-T cells were used. Considering the disease, one study included DLBCL patients, while the rest included ALL patients. Both CRS and neurotoxicity were observed more frequently in patients treated with CAR-T cells. “

Round 2

Reviewer 3 Report

The paper has been improved in terms of language and style. However, it is very extensive and difficult to read for the target population, the 'on-duty doctor'. The list of citations is also very extensive. The literature review on the effectiveness of CAR T cells and bispecific antibodies seems dispensable.

Author Response

Thank you very much for an important feed-back. We extended the manuscript and described in detail the management of the critically ill hematology patient following the suggestions of the other reviewers. Still, we consider that the present manuscript is much better, as it addresses all major problems that hematology patients experience in the clinic. We insisted on including the data on CAR T cells and bispecific antibodies as these emerging therapies are currently under investigation in dozens of phase I-III clinical trials and will change the landscape of immunotherapy and stem cell transplantation in the following 3-5 years. By keeping the data on modern immunotherapy, the manuscript bring forward modern and state-of-the-art data, of interest in hematology wards of 2019 and 2020.